# Automated Filling of Dry Micron-Sized Particles into Micro Mold Pattern within Planar Substrates for the Fabrication of Powder-Based 3D Microstructures

**DOI:** 10.3390/mi12101176

**Published:** 2021-09-29

**Authors:** Cris Kostmann, Thomas Lisec, Mani Teja Bodduluri, Olaf Andersen

**Affiliations:** 1Fraunhofer Institute for Manufacturing Technology and Advanced Materials IFAM, Winterbergstrasse 28, 01277 Dresden, Germany; cris.kostmann@ifam-dd.fraunhofer.de (C.K.); olaf.andersen@ifam-dd.fraunhofer.de (O.A.); 2Fraunhofer Institute for Silicon Technology, Fraunhoferstrasse 1, 25524 Itzehoe, Germany; mani.teja.bodduluri@isit.fraunhofer.de

**Keywords:** 3D microstructures, MEMS, particle packing, ultrasound, NdFeB micromagnets

## Abstract

Powder-based techniques are gaining increasing interest for the fabrication of microstructures on planar substrates. A typical approach comprises the filling of a mold pattern with micron-sized particles of the desired material, and their fixation there. Commonly powder-loaded pastes or inks are filled into the molds. To meet the smallest dimensions and highest filling factors, the utilization of dry powder as the raw material is more beneficial. However, an appropriate automated technique for filling a micro mold pattern with dry micron-sized particles is missing up to now. This paper presents a corresponding approach based on the superimposition of high- and low-frequency oscillations for particle mobilization. Rubber balls are utilized to achieve dense packing. For verification, micromagnets are created from 5 µm NdFeB powder on 8” Si substrates, using the novel automated mold filling technique, as well as an existing manual one. Subsequent atomic layer deposition is utilized to agglomerate the loose NdFeB particles into rigid microstructures. The magnetic properties and inner structure of the NdFeB micromagnets are investigated. It is shown that the novel automated technique outperforms the manual one in major terms.

## 1. Introduction

Powder-based techniques are of particular interest for microelectromechanical sys-tems (MEMS), since they allow the creation of three-dimensional (3D) structures with di-mensions between tens and hundreds of microns, from a broad range of materials. But popular additive manufacturing techniques like selective laser beam sintering [1] or melt-ing [2] are rather not suitable in combination with silicon or glass substrates as commonly used for MEMS. Techniques like ink-jet printing are too time consuming since 3D struc-tures are created layer-by-layer in multiple repeated sequences [3]. Alternatively, the filling of a mold pattern within a planar substrate with particle-loaded inks or pastes by spin coating, squeegee coating, or molding and subsequent solidification of the matrix material by heat treatments, for example, is widely applied [4]. However, for molds with dimensions as mentioned above, those classic techniques are less suitable. At the smallest scale, 3D structures with the maximum particle content are hardly achievable in this way, since the viscosity of the utilized ink or paste quickly rises with the particle loading. Dry pressing of micron-sized powder without any additives is a promising alternative. For example, the smallest, densely packed micromagnets have been fabricated from NdFeB powder in such a way [5,6]. Yet, this approach entails additional effort. As depicted in Figure 1, after filling the dry powder into a micro mold pattern (step I), the loose particles must be fixated permanently using a dedicated technique (step II). In [5], for that purpose, polyimide is spin-coated onto the substrate to keep the particles within the molds, and, in [6], parylene is deposited on top instead. However, both techniques are rather inconvenient for production needs. As recently demonstrated, atomic layer deposition is well suited to agglomerate dry particles within micro molds on the wafer level into 3D structures with high mechanical and thermal stability [7]. Thus, for step II of the procedure in Figure 1, an appropriate technique that is suitable for the mass production of MEMS is available. However, an adequate method for step I, the filling of micro molds with dry powder, is still missed. This strongly impedes the utilization of the fabrication scheme in Figure 1 in practice. 

Commonly, the dry filling of particles into micro molds within silicon substrates is performed manually, using simple implements [5,6,7]. First, a certain amount of powder is poured onto the surface. Then, the powder is spread over the micro mold pattern by wiping a squeegee (doctor blade) across the substrate. To reliably move the particles into the micro molds, and to ensure dense packing, sufficient downward pressure must be applied. This downward pressure is not provided by common doctor blading tools. Usually, doctor blading is applied in combination with inks or slurries to fabricate homogenous films on flat surfaces, or for tape casting (see Figure 2a,b). In metal additive manufacturing, dry powder is raked across the working bed in each cycle (see Figure 2c). In all those cases, skewing of the blade is sufficient to ensure full functionality. An additional downward force is not needed.

In this work, a technique for the automated filling of micro molds within Si substrates, with dry micron-sized powder, is presented for the first time. The novel technique is evaluated for ceramic, as well as for metallic powder. In order to verify it quantitatively, and compare it with the currently utilized manual technique, micromagnets are fabricated on the wafer level from NdFeB powder. Their magnetic properties and inner build-up are evaluated. It is shown that the novel automated technique outperforms the manual one in major aspects. 

## 2. Materials and Methods

In all experiments 8” silicon substrates were used. A mold pattern was created by deep reactive ion etching (DRIE) of the 725-µm-thick Si substrate on a SPTS Pegasus using a resist mask. Figure 3a shows the designs, which were examined with respect to the mold filling quality. R1000 and G2 should disclose the impact of the feature size on uniformity and density of the filling. M5 should indicate the impact of the feature size on the filling level. Please note that the depth of all those structures is different. As depicted by the graph in Figure 3b, the DRIE depth strongly depends on the feature size. For R1000, depth values of about 540 µm were measured. The pixels of G2 are about 375 µm in depth. The depth values of the trenches of design M5 correspond to the first 6 data points of the graph in Figure 3b.

Table 1 summarizes the properties of the powders utilized during the development of the automated mold filling technique. The feedstock was analyzed in terms of powder size distribution, bulk density, tap density and adhesion to a polished Si surface. Figure 4 presents optical micrographs of the 10 µm Al_2_O_3_ powder and the NdFeB powder. Figure 5 shows the exemplarily particle distribution of the used NdFeB powder containing a particle spectrum in the range from 2 µm to approx. 18 µm.

To identify suitable process settings for the novel automated mold filling technique, firstly Si substrates with the mold pattern shown in Figure 3 were processed using the Al_2_O_3_ powders in Table 1. However, fine tuning or quantitative assessment are not possible with those powders due to the lack of easily measurable properties correlated to packing density and quality of the mold filling. For that, micromagnets were fabricated from the NdFeB powder following the procedure in Figure 1. After filling the micro molds with the dry powder in accordance to step I in Figure 1, either using the novel automated technique or the manual procedure pre-existing at Fraunhofer ISIT, the NdFeB particles were agglomerated to rigid 3D microstructures in accordance to step II in Figure 1 by 75 nm ALD Al_2_O_3_ deposited at 75 °C using TMA and water as precursors. After dicing a selected number of Si chips with embedded micromagnets of R1000 and G2, design was picked from each substrate and measured on a LakeShore 7400 vibrating sample magnetometer (VSM) to obtain the magnetic properties of the integrated micromagnets. Figure 6 exemplarily shows typical hysteresis curves for both designs. Of particular interest are the remanence B_r_ and the intrinsic coercivity H_ci_. Since B_r_ depends on the amount of magnetic material, this value represents a measure for the mold filling quality. H_ci_ is an intrinsic material property and does not depend on the volume of a magnet. Deviations would indicate a degradation of the magnetic material due to the applied procedures. 

On chips of design M5 surface profile, scans of the embedded micromagnets were obtained using a Dektak 3ST profilometer. The scans reflect the impact of the filling procedure on the filling level as function of the mold width. More details regarding substrate processing or properties of the NdFeB-based micromagnets can be found in [8,9]. 

For all tests, NdFeB powder MQFP-B+(D50 = 5 µm)-10215-089 from Magnequench has been utilized. For B_r_ and H_ci_ as typical values 865–895 mT and 701–836 kA/m are specified in the data sheet.

## 3. Mold Filling Procedures

### 3.1. Manual Mold Filling

Figure 7 depicts the manual mold filling procedure, as commonly applied at Fraunhofer ISIT and as used for comparison with the novel automated mold filling described within this work. In the first step, sufficient powder is poured onto the substrate (Figure 7a). Then, the powder is uniformly spread over the surface of the substrate using an applicator with a rubber blade (Figure 7b). In the next step, the powder is pressed into the micro mold pattern using a rubber roll to increase the fill factor (Figure 7c). Finally, the excess powder is wiped away carefully using the applicator with a rubber blade once more (Figure 7d). The quality and reproducibility of manual mold filling, to a certain extent, depends on the skills of the operator. Nevertheless, it is well reproducible, showing standard deviations between 4% and 8% over a 200 mm Si substrate [9].

### 3.2. Automated Mold Filling

A prerequisite for the automated filling of the micro cavities is the homogeneous distribution of the dry powder over the entire surface of the substrate. In this work, a low-frequency motion is combined with a high-frequency (ultrasound) vibration to induce particle motion. As it has been found, low-frequency shaking alone is not sufficient. The fine powder is not normally pourable and falls into the cavities without filling them completely. After a certain amount time of operation, the movement of the powder over the wafer surface stops, although it is still being “shaken”. The addition of ultrasound reduces the friction in such a manner that the particles continue to flow over the cavities. However, solely ultrasound does not cause powder movement across the wafer surface. Only the combination of both low-frequency shaking and high-frequency ultrasound ensures continuous powder movement, and, thus, complete mold filling. Please note that the ultrasound partially resolves smaller agglomerates into the particles of primary size. However, this does not lead to further destruction of the primary particles. Therefore, the particle size distribution, as shown for the NdFeB powder in Figure 5, remains unchanged.

Figure 8 illustrates the mold filling setup. It is based on a sieve shaker EML200 from Haver & Boecker providing the three-dimensional low-frequency motion. The Si substrate is vacuum clamped on a custom-built chuck with an integrated UP200St ultrasound transducer and matching generator from Hielscher. A custom-built ring-shaped fixture, matching the fastening system of the shaker, is used to connect the vacuum chuck rigidly with the machine body. The inner edge of this ring-shaped fixture overlaps the substrate by a few millimeters, so that the powder is kept within the substrate area during operation.

For mold filling, first a certain amount of dry powder is poured onto the substrate. Then, the shaker and ultrasound transducer are switched on. The superimposition of the three-dimensional low-frequency shaking with the high-frequency ultrasound forces the micron-sized particles reliably upon a rotational movement over the substrate. Figure 9 shows exemplarily photographs of the chuck assembly with the mounted substrate in the initial state after the dry NdFeB powder has been poured onto it, and after completed mold filling. Within a short time, a wedge-shaped powder bed is formed, which travels clockwise over the substrate. In such a way, particles are reliably provided to all the micro molds. The overall process time is about 10 min.

As already mentioned in the introduction, sufficient downward pressure must be applied to the particles to move them into the micro molds and to ensure a high filling density. During manual mold filling, a rubber roll is used for this (see Figure 7c). In the case of automated mold filling, rubber balls support the compaction of the deposits in the cavities. Please note that the utilization of rubber balls with either the low- or the high-frequency motion does not work properly; consistent filling is only achieved in combination with both motions. In addition, the rubber balls further resolve the agglomerates. The latter is especially required when processing irregular particles. However, the disintegration of the agglomerates does not cause destruction of the particles. The rubber balls are placed on the substrate before pouring the dry powder, and they should cover the entire surface. Accelerated by the applied vibrations, the rubber balls bounce and rotate around their axis, and simultaneously move together with the particle bed, clockwise over the substrate. Figure 10a presents a photograph of the chuck assembly with the clamped Si substrate in the initial state, after the rubber balls were placed and dry NdFeB powder was poured. In Figure 10b, the same substrate is shown after the mold filling was completed and the rubber balls were removed, by tilting the chuck assembly. As can be observed, the excess powder is significantly grouted. 

After mold filling, the excess powder must be removed from the substrate surface. For that purpose, the vacuum chuck with the substrate is taken from the shaker setup. Optionally, the rubber balls are removed as well. Then, the chuck is mounted on a squeegee setup comprising a CX-4 automated film applicator from MTV Messtechnik, with vertically adjustable polymer blade and custom-built adapter for precise fixation. Figure 11a shows a corresponding photograph. To avoid the generation of filling defects within the micro molds, it is essential to remove the excess powder step by step. Therefore, the polymer blade is moved repeatedly in several cycles, with a decreasing gap over the substrate. Figure 11b presents a photograph of the same substrate as in Figure 9b, after the final squeegee cycle with a 5 µm gap. 

Figure 12 summarizes the automated mold filling procedure schematically. In general, the settings of both the shaker and squeegee setups must be optimized for each individual type of powder, even if it consists of the same material. Particle shape, size, and size distribution play an important role. For example, both Al_2_O_3_ powders in Table 1 are irregular in shape and therefore tend to form agglomerates, in the dry state especially. This makes it difficult to distribute the powder uniformly over the substrate surface. Overfilling of the micro molds and bridging between neighboring ones is observed. The metallic NdFeB powder contains a small fraction of spherical particles (see Figure 5). This means reduced agglomerate formation and even more distribution over the full surface of the substrate. The narrow particle size distribution, on the other hand, resulted in lower compaction and, due to that, in a lower packing density.

Table 2 and Table 3 summarize the basic process settings, as applied in the case of the powders in Table 1. Please note that they can vary, even for the same powder, depending on, for example, the powder batch or the humidity of the ambient atmosphere. 

## 4. Comparison of Automated and Manual Mold Filling

### 4.1. Optical Appearance of Substrate Surface

During mold filling, as described above, the particles unavoidably remain on the substrate surface around the micro molds. The following agglomeration of the powder within the micro molds, by ALD (step II in Figure 1), fixates those particles quite rigidly to the surface. However, for further processing of the substrate, smooth and clean surfaces are often mandatory. Suitable procedures for appropriate surface conditioning are already described in [3,4]. Nevertheless, the initial surface contamination, due to the mold filling procedure, should be as low as possible. As Figure 13 indicates, automated filling results in markedly less particles on the surface compared to the manual procedure.

### 4.2. Magnetic Properties of Integrated Micromagnets

Figure 14 summarizes the mean value (MV) and the standard deviation (SD) of the remanence B_r_ of the micromagnets of designs R1000 and G2 as a function of the applied mold filling procedure. All the values were calculated from 16 chips of the corresponding design which were picked within the central area of the substrate and measured on the VSM. Automated mold filling has been performed using the settings in Table 2 and Table 3, and manual mold filling was performed as illustrated in Figure 7.

Both the manual as well as the automated procedure with compaction result in higher B_r_ values than without. In the case of manual mold filling, the difference between “with” and “without” is rather small, since this procedure already comprises considerable downward forces, even without compaction. Besides, the applied compaction seems to be less effective in the case of small-sized structures. For G2, consisting of 125 µm × 125 µm pixels, the increase in B_r_ is markedly lower compared to R1000, which is a single structure that is 2000 µm × 2000 µm in size. For automated filling without compaction, the lowest B_r_ values are measured. In contrast to the automated filling without compaction, with compaction yields the highest B_r_ values. Conspicuously for design G2, obtained by automated filling without compaction, the SD of B_r_ is about three times higher compared to all the other variations.

To visualize the packing of the NdFeB-based 3D microstructures, the silicon surrounding them was largely removed on the chip level, by selective isotropic etching in the XeF_2_ gas phase. Figure 15 illustrates this procedure.

Figure 16 presents SEM micrographs of the bottom side of the micromagnets of design G2, fabricated using different filling procedures after Si removal in the XeF_2_ gas phase, as illustrated in Figure 15. Due to an acceleration voltage of 20 kV, the Al_2_O_3_ ALD layer, covering the former mold bottom, is penetrated, and the agglomerated particles beneath can be observed. Please note that for all the samples, the same NdFeB powder from the same manufacturer lot were utilized. The visual differences are caused by the different packing methods. From the appearance of the 3D microstructures, it could be concluded that the highest packing density is achieved by automated filling with compaction. Automated filling without compaction results in a somewhat lower density. Manual filling with compaction yields, by far, the lowest packing density. This is in contradiction to the results of the VSM measurements (see Figure 14a). By far, the lowest remanence B_r_ was measured for automated filling without compaction. At the same time, for manual and automated filling with compaction, the B_r_ values do not differ as much as Figure 16a,c suggest. 

Those discrepancies can be explained by the presence of voids within the micromagnets. Depending on the mold filling procedure, different types of voids were observed. Figure 17 presents the typical voids appearing in the case of automated mold filling without compaction. In Figure 17a, a small void can be observed at the mold bottom of a sample of the R1000 design. Small-sized structures, such as the pixels within an array of G2 design, can remain completely empty (see Figure 17b). Accidentally, this second type of void also explains the exceptionally high standard deviation of B_r_ for G2 in this case (Figure 14b).

Micromagnets that are fabricated by manual filling with compaction exhibit a third void type. Here, large-sized voids can be found at the mold bottom, along the edges and in the corners. Figure 18 presents corresponding SEM micrographs. 

The formation mechanisms of the different voids are not yet fully evaluated. However, the small voids that were observed after automated filling without compaction (Figure 17a) seem to be associated with the presence of agglomerates within the 3D microstructures. As the cross-sectional polish in Figure 19a illustrates, many small voids can be found between neighboring agglomerates within the bulk of the micromagnet. Due to this fact, for automated filling without compaction, exceptionally low B_r_ values are measured for both the G2 and R100 designs (Figure 14a), although the packing density appears to be rather high, visually (Figure 16b). It is assumed that the agglomerates are already present in the raw NdFeB powder, before the filling procedure. Consequently, the large voids, as shown in Figure 17b, most probably occur when a small-sized structure is blocked by an agglomerate within its upper part. The ultrasound applied during mold filling is not sufficient to resolve those agglomerates reliably. In contrast to that, no agglomerates are found if automated filling was performed with compaction (Figure 19b). Accordingly, for this filling method, the highest B_r_ values are measured. 

Due to the voids, as shown in Figure 18, the B_r_ values for manual filling with compaction are about 10% lower compared to automated filling with compaction (Figure 14a). Obviously, the rubber roll that is utilized for compaction provides enough force to resolve the agglomerates and to avoid the corresponding small voids within the bulk. However, the corners and edges at the mold bottom cannot be reliably filled in this way. This seems to be possible only if adequate compaction is combined with ultrasound.

Please note that manual filling without compaction has not been investigated. As Figure 7 illustrates, manual filling always comprises compaction, to a considerable extent. Therefore, the difference between the B_r_ values for manual filling with and without compaction is not as high as in the case of the corresponding automated procedures (Figure 14a). However, with compaction, the filling procedure is better defined and less dependent on the skills of the operator.

The intrinsic coercivity H_ci_ is a material parameter and does not depend on the density or volume of a magnet, and should be not affected by the filling procedure. Accordingly, no dependency of the MV and SD of H_ci_ from the filling method can be derived from the plots in Figure 20.

### 4.3. Mold Filling Height

Figure 21 presents line scans over the surface of the NdFeB-based microstructures of the M5 design, obtained with a Dektak profilometer. In the case of manual filling with compaction, the surface profile of the 3D microstructures shows a meniscus-type shape, which increases with its width. For example, the micromagnet within the 500-µm-wide trench is in its center up to 50 µm below the surface. In the case of automated filling, the dependency of the surface profile from the trench width is less pronounced. With compaction, this dependency is lower by factors than without compaction. 

## 5. Discussion

An automated technique for the filling of micro molds within planar substrates, with dry micron-sized powder, has been developed. In combination with particle agglomeration by ALD, it allows the generation of rigid and durable 3D microstructures for MEMS, from a broad range of materials on common Si substrates. For the first time, a fabrication scheme, as depicted in Figure 1, is available for mass production.

It is shown that the novel automated mold filling technique outperforms an existing manual procedure in major terms. Compared to the manual method, the automated filling method leads to better reproducibility, while reducing the process time. A comparison of the remanence B_r_ of micromagnets obtained from NdFeB powder revealed that the 3D microstructures that are created by automated mold filling with compaction show a higher packing density and less packing defects. The contamination of the substrate surface with particles is less pronounced. Even in this early stage of development, automated filling with compaction ensures the most uniform surface profile, with the lowest dependency on the mold dimensions. With a standard deviation of B_r_ up to 6%, a good homogeneity over a 200 mm Si substrate is demonstrated. From the standard deviation of the intrinsic coercivity H_ci_, remaining well below 1%, it can be concluded that the NdFeB powder does not degrade, due to the treatments applied during automated mold filling.

Further development will mainly focus on the improvement of the described setups, as well as the optimization of the whole workflow shown in Figure 12. An indispensable prerequisite for that is the availability of dedicated characterization tools for measurements on the wafer level, regarding the magnetic properties of integrated NdFeB-based micromagnets especially. Corresponding investigations are ongoing.

## Figures and Tables

**Figure 1 micromachines-12-01176-f001:**
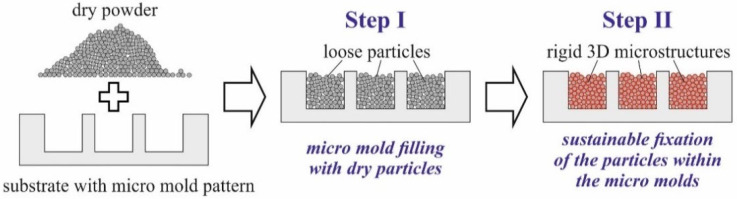
The fabrication of rigid 3D microstructures from dry micron-sized powder on a planar substrate with micro mold pattern basically comprises two steps. Step I: fill spreading of dry powder into the micro molds. Step II: permanent fixation of the loose particles within the micro molds.

**Figure 2 micromachines-12-01176-f002:**
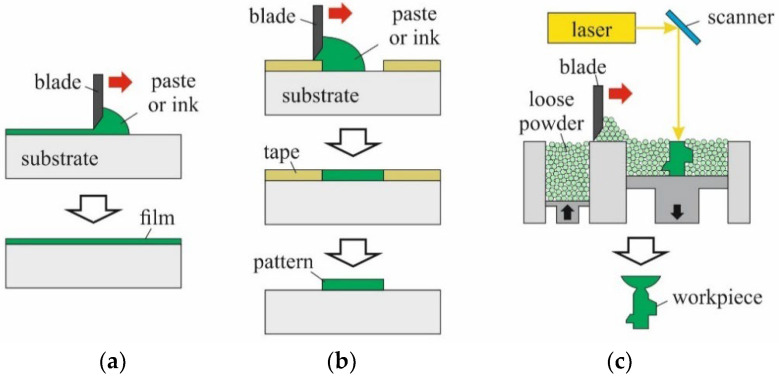
Schematic illustration of common applications of doctor blading: (**a**) thin film coating, (**b**) tape casting and (**c**) additive manufacturing of workpieces.

**Figure 3 micromachines-12-01176-f003:**
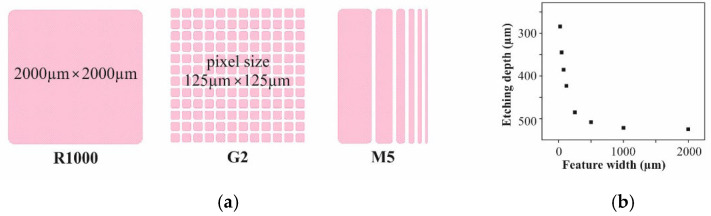
(**a**) Layout of the micro mold pattern used for investigations within this work. M5 comprises an array of 2-mm-long trenches with 25, 50, 75, 125, 250 and 500 µm width (from left to the right) and 60 µm spacing. (**b**) Dependency of the DRIE depth from the width of the etched feature.

**Figure 4 micromachines-12-01176-f004:**
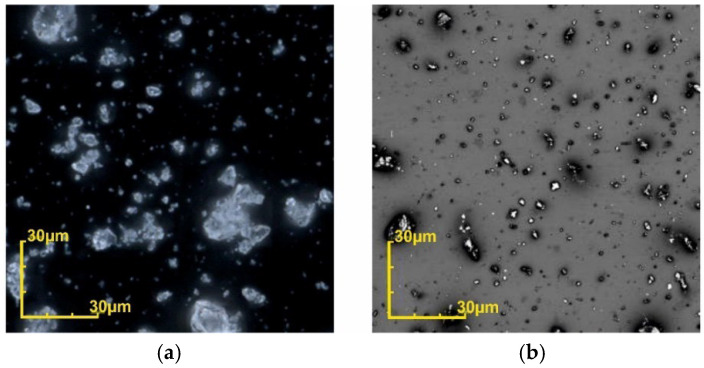
Optical micrographs of (**a**) the 10 µm Almatis Al_2_O_3_ and (**b**) the NdFeB powder.

**Figure 5 micromachines-12-01176-f005:**
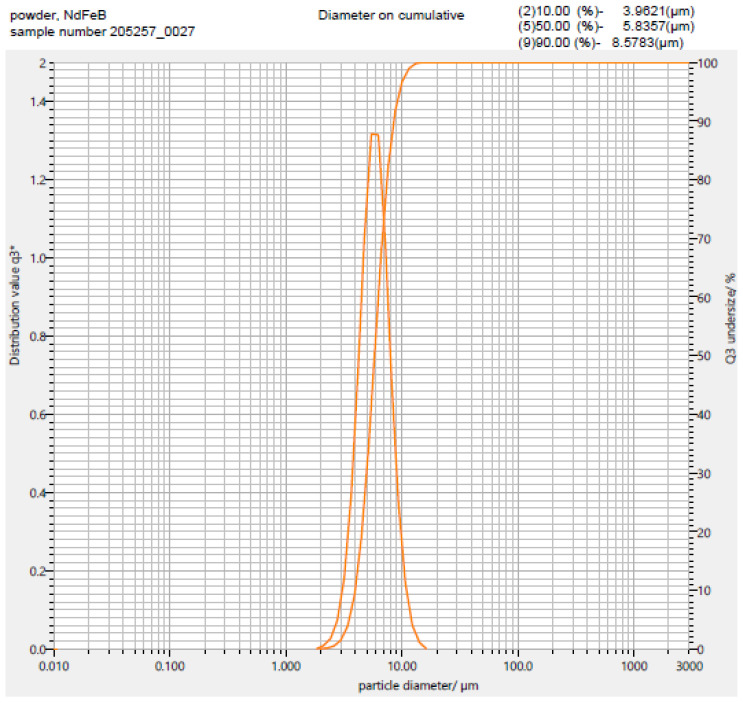
Particle size distribution of the utilized NdFeB power with a d_50_ = 5 µm (Magnequench) obtained using an LA950 laser-scattering particle analyzer from Retsch.

**Figure 6 micromachines-12-01176-f006:**
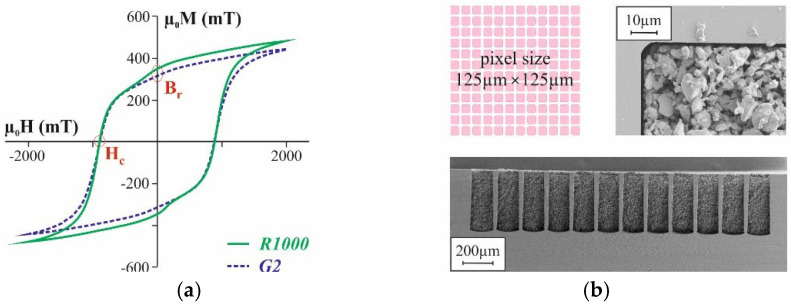
(**a**) Typical magnetization curves for NdFeB micromagnets of design R1000 and G2 as fabricated within this work. (**b**) Layout of design G2 and SEM micrographs of one pixel from the top as well as a diced through chip. The NdFeB particles are agglomerated by ALD so that the parts of the porous structures remaining after dicing do not fall out of the cut through micro molds.

**Figure 7 micromachines-12-01176-f007:**
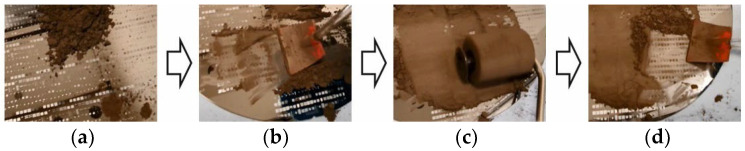
Illustration of the manual filling of a micro mold pattern on an 8” silicon substrate with NdFeB powder. The procedure comprises up to four steps, as follows: (**a**) pouring of the powder onto the substrate; (**b**) spreading of the powder across the substrate; (**c**) optional pressing of the powder into the micro molds for additional compaction to increase the fill factor; (**d**) wiping away the excess powder.

**Figure 8 micromachines-12-01176-f008:**
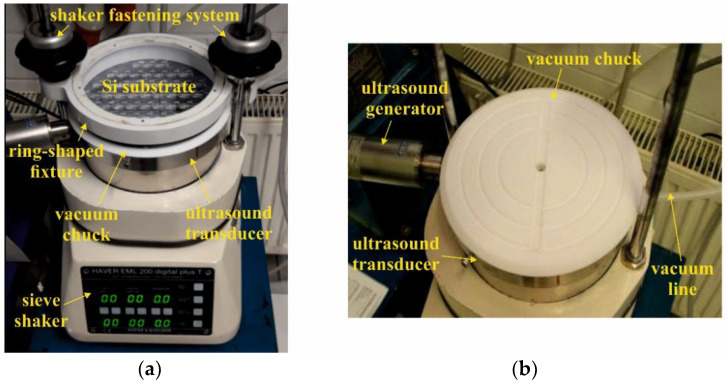
(**a**) Setup for automated micro mold filling consisting of a sieve shaker and a custom-built clamping unit with mounted substrate. The clamping unit comprises a vacuum chuck with integrated ultrasound transducer and a ring-shaped cover. In (**b**) the clamping unit is shown without cover and substrate.

**Figure 9 micromachines-12-01176-f009:**
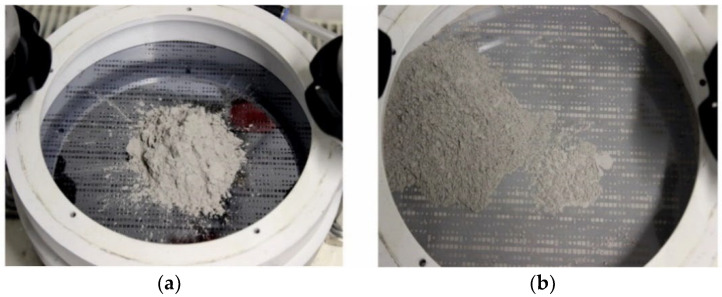
Photographs of the chuck assembly with mounted Si substrate (**a**) in the initial state after pouring of dry NdFeB powder within its center and (**b**) after completed mold filling.

**Figure 10 micromachines-12-01176-f010:**
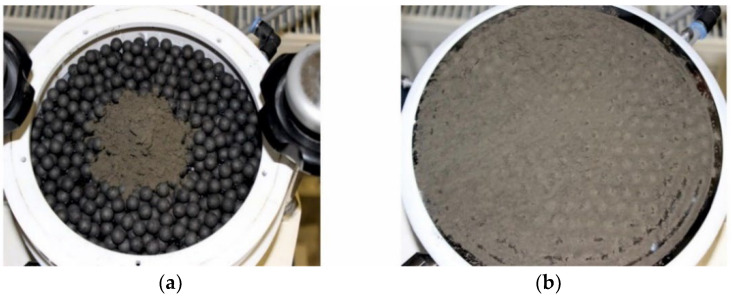
(**a**) Photograph of the chuck assembly with mounted substrate after placing rubber balls on its surface and pouring of NdFeB powder. In (**b**) the same substrate is shown after completed mold filling, dismounting of the vacuum chuck from the shaker setup and rubber ball removal by tilting the chuck assembly.

**Figure 11 micromachines-12-01176-f011:**
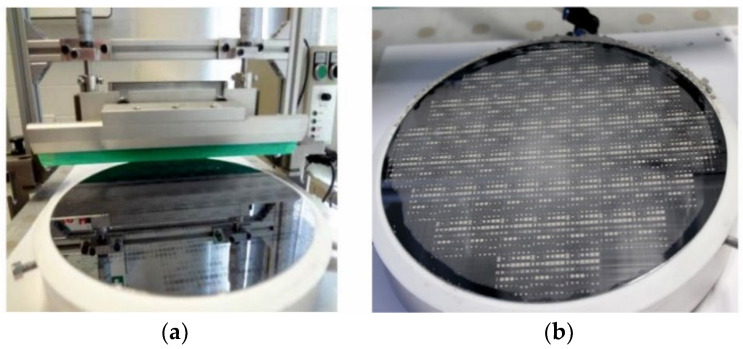
(**a**) Squeegee setup for removal of excess particles comprising a film applicator with a custom-built adapter for precise fixation of the vacuum chuck with the Si substrate. Please note, the micro molds of the substrate shown here are empty. The photograph in (**b**) presents a substrate processed using NdFeB powder after the final squeegee cycle with 5 µm gap.

**Figure 12 micromachines-12-01176-f012:**
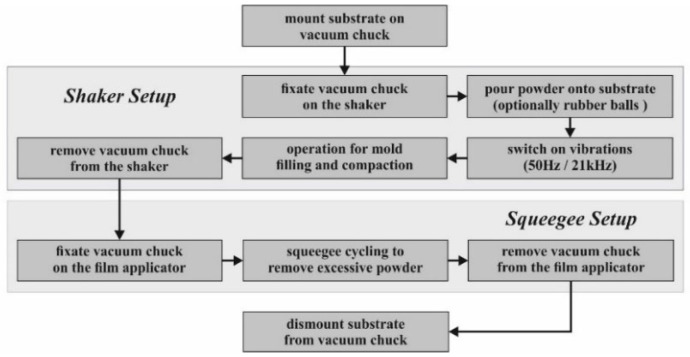
Schematic workflow of the automated mold filling procedure as described in this work.

**Figure 13 micromachines-12-01176-f013:**
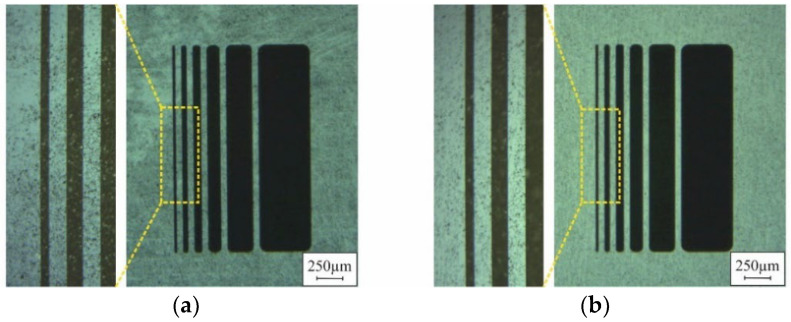
Photographs of chips of M5 design on substrates after (**a**) manual and (**b**) automated mold filling of NdFeB particles and subsequent agglomeration by ALD.

**Figure 14 micromachines-12-01176-f014:**
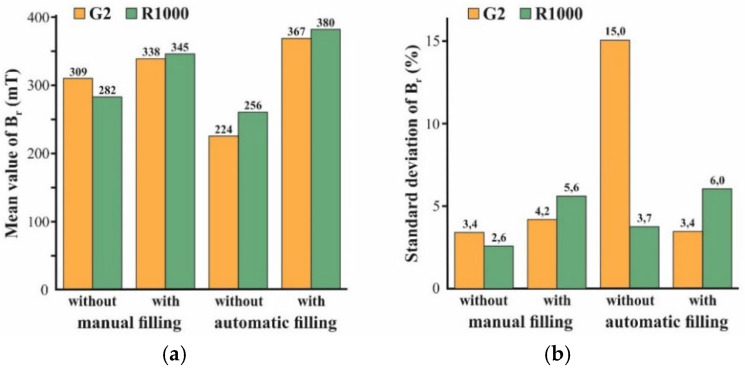
Mean values (**a**) and standard deviations (**b**) of the remanence B_r_ for micromagnets of designs G2 and R1000 from substrates that were processed using manual or automated mold filling with or without compaction. Please note that from each substrate 16 chips of each design were picked and measured to obtain the shown values.

**Figure 15 micromachines-12-01176-f015:**
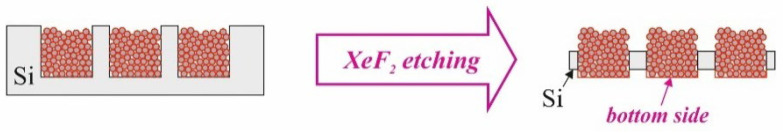
Schematic illustration of the etching in XeF_2_ gas phase applied to remove the silicon substrate surrounding the NdFeB-based micromagnets and to expose their bottom side.

**Figure 16 micromachines-12-01176-f016:**
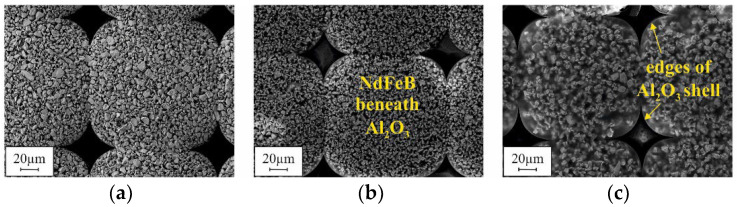
SEM micrographs of the bottom side of micromagnets of design G2 fabricated using (**a**) automated filling with compaction, (**b**) automated filling without compaction and (**c**) manual filling with compaction. The micrographs were obtained with an acceleration voltage of 20 kV. The surrounding silicon has been removed in XeF_2_ gas phase as illustrated in Figure 15.

**Figure 17 micromachines-12-01176-f017:**
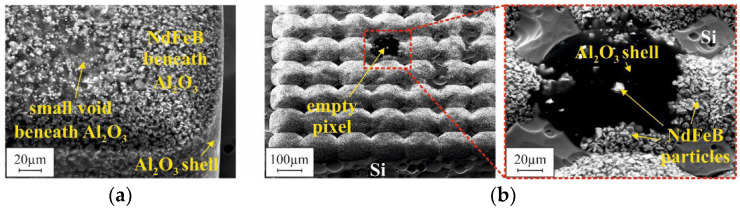
SEM micrographs of the bottom side of micromagnets, obtained by automated mold filling, after removal of the surrounding silicon in XeF_2_ gas phase as illustrated in Figure 15. (**a**) Structure of R1000 design showing a small void. (**b**) Structure of G2 design exhibiting a pixel that is empty in the lower part and detailed view of this pixel (area within the red square). All micrographs were made with an acceleration voltage of 20 kV.

**Figure 18 micromachines-12-01176-f018:**
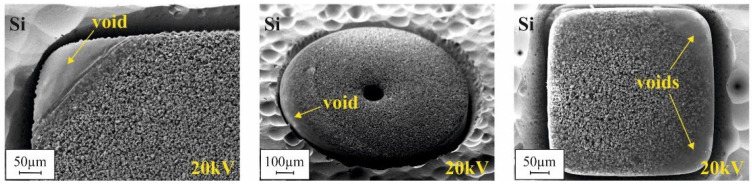
SEM micrographs of the bottom side of various micromagnets fabricated utilizing manual filling with compaction after removal of the surrounding silicon in XeF2 gas phase as illustrated in Figure 15. All micrographs were obtained with an acceleration voltage of 20 kV.

**Figure 19 micromachines-12-01176-f019:**
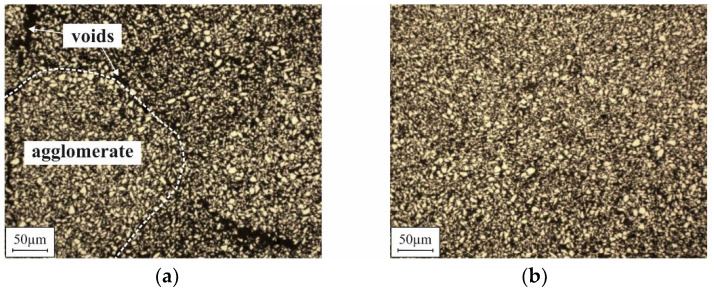
Images of cross-sectional polishes through 3D microstructures fabricated using (**a**) automated filling without compaction and (**b**) automated filling with compaction.

**Figure 20 micromachines-12-01176-f020:**
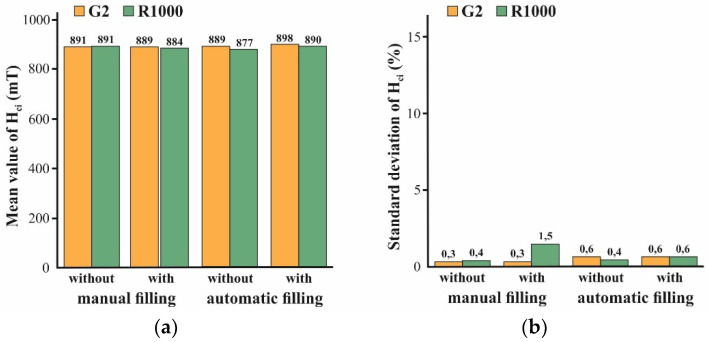
Mean values (**a**) and standard deviations (**b**) of the intrinsic coercivity H_ci_ for micromagnets of designs G2 and R1000 from substrates that were processed using manual or automated mold filling with or without compaction. From each substrate, 16 chips of each design were picked and measured. The corresponding B_r_ values are summarized in Figure 14.

**Figure 21 micromachines-12-01176-f021:**
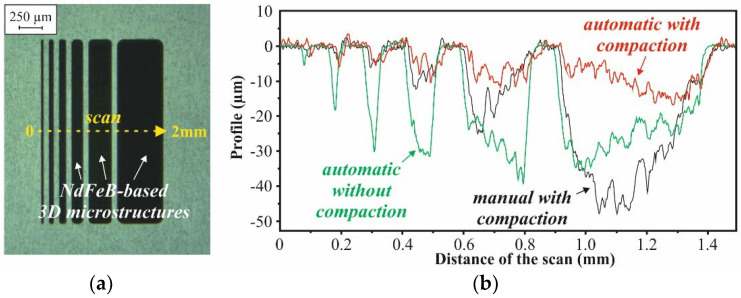
(**a**) Photograph of a chip of design M5 with an array of 2-mm-long 3D microstructures with 25, 50, 75, 125, 250 and 500 µm width (from left to the right) and (**b**) exemplary profilometer scans over chips from substrates processed using different mold filling procedures. The gap between neighboring structures is 60 µm.

**Table 1 micromachines-12-01176-t001:** Particle size distribution and densities of different powders used for filling experiments.

Powder	Size Distribution (µm)				Apparent Density (g/cm^3^)	Tap Density (g/cm^3^)	Surface Adhesion Characterization
	d_10_	d_50_	d_90_	std. dev.			
2.2 µm Al_2_O_3_ (Almatis)	0.7692	2.3773	5.6045	2.0045	0.9327	1.3324	intermittent film
10 µm Al_2_O_3_ (Almatis)	1.6469	7.1142	15.0066	10.8078	1.1967	1.6894	intermittent film
5 µm NdFeB Magnequench)	3.9621	5.8357	8.5783	1.8556	1.8817	2.6881	closed film with selective clustering

**Table 2 micromachines-12-01176-t002:** Main settings of the shaker setup for processing of Si substrates with the powders in Table 1.

Rubber Balls (Ø 10 mm)	Amount of Powder	Sieve Shaker		Ultrasound Transducer		Duration
		Frequency	Amplitude	Frequency	Amplitude	
no	50 g	50 Hz	0.5–2 mm	20–60 kHz	1–4 µm	640 s
yes						180 s

**Table 3 micromachines-12-01176-t003:** Main settings of the squeegee setup for processing of Si substrates with the powders in Table 1.

Blade Speed	Gap 1st Cycle	Gap 2nd Cycle	Gap 3rd Cycle
80 mm/s	200 µm	50 µm	5 µm

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
