# Peer review of "Automated Filling of Dry Micron-Sized Particles into Micro Mold Pattern within Planar Substrates for the Fabrication of Powder-Based 3D Microstructures"

_micromachines, 2021, doi:10.3390/mi12101176_

Round 1
Reviewer 1 Report
This paper reported an automated method to fill microparticles to mold pattern. This method combines low frequency shaking and ultrasound vibration to achieve better packing density. Experiments and details were well presented, but there are some major concerns that need to be addressed.
- Authors claimed that the new automated filling method can overperform manual filling. This statement does not stand well. From Fig 14, it shows that automated filling method has to use rubber ball for compaction to achieve better filling performance. Is ultrasound or shaking necessary to achieve better performance? What are the functions of shaking and ultrasound? Probably shaking + rubber ball can give you better performance already. This is a major question for this paper. Suggest to add some data with conditions below: 1. Shaking only + rubber ball, no ultrasound; 2. Ultrasound +rubber ball, no shaking
- There are also some minor comments listed below:
- When you did VSM measurement, is Si substrate also included in the sample? How to control the amount of Si in samples?
- There are some typos, please check thorough the manuscript. e.g line 252, “Then” should be “than”
- Fig 3a is confusing since only R1000, G2, M5 pattern are characterized, think to make manuscript more concise, can delete fig3a. Otherwise, need to explain all the other patterns.
- In Fig16, particle size look different in a,b, why?
- Fig18b, is there a void at center? Does it have different void mechanism? Is it from a different mold? Image is presented in the manuscript but it was not explained.
- Fig20, why there is no data for manual filling without compaction?
Author Response
- Authors claimed that the new automated filling method can overperform manual filling. This statement does not stand well. From Fig 14, it shows that automated filling method has to use rubber ball for compaction to achieve better filling performance. Is ultrasound or shaking necessary to achieve better performance? What are the functions of shaking and ultrasound? Probably shaking + rubber ball can give you better performance already. This is a major question for this paper. Suggest to add some data with conditions below: 1. Shaking only + rubber ball, no ultrasound; 2. Ultrasound +rubber ball, no shaking
We added more inforamtion regadring the filling process in chapter 3.2. The automated filling is working reproducilbe only if shaker motion, utrasound and rubber balls are utilized. Therefore data on selected combinations (for example "shaking only + rubber balls") cannot be provided.
- There are also some minor comments listed below:
- When you did VSM measurement, is Si substrate also included in the sample? How to control the amount of Si in samples?
Since Si is not magnetic it not contributes to the B-H curves. Control of the amount of Si in the samples in not necessary. However, the sample size is always the same since the chips are obtained by high precision dicing.
- There are some typos, please check thorough the manuscript. e.g line 252, “Then” should be “than”
Thank you for the comment. Correction has been made.
- Fig 3a is confusing since only R1000, G2, M5 pattern are characterized, think to make manuscript more concise, can delete fig3a. Otherwise, need to explain all the other patterns.
Done
- In Fig16, particle size look different in a,b, why?
The particle size is not different, but it indeed looks like that. For all tests the sameNdFeB powder was used.
- Fig18b, is there a void at center? Does it have different void mechanism? Is it from a different mold? Image is presented in the manuscript but it was not explained.
Explanations have been added in the corresponding text.
- Fig20, why there is no data for manual filling without compaction?
Manual filling always includes significant compaction, see Figure 7. It is rather not possible to avoid that. Correspondingly the difference between both manual procedures ist low, i.e. the corresponding Br values differs not as much. Main issue is that without the additional compaction the procedure is less reproducible and depends more on the operator.
Reviewer 2 Report
The present manuscript deals with automated filling of dry micron-sized particles into micro mold pattern within planar substrates for the fabrication of powder-based 3D microstructures. This paper presents a corresponding approach based on superimposition of high- and low-frequency oscillations for particle mobilization. For verification micromagnets are created from 5µm NdFeB powder on 8” Si substrates, The overall quality and construction are well matched, and the experimental results supported the authors' claims. In addition, there are few issues needed to be addressed, and therefore, minor revision is required for publication in microelectromech.
1) Particle size distribution of NdFeB powder should be provided after completed mold filling.
2) The presence of NdFeB particles should be shown in Figure 17.
3) The labels in Figure 20 need to be placed more reasonably.
4) The intrinsic coercitivity Hci of NbFeB designed by M5 should be provided such as G2 and R1000.
5) The main purpose of the application should be given for the readers to better understand the work.
Author Response
1) Particle size distribution of NdFeB powder should be provided after completed mold filling.
The particle size distribution not changes. Corresponding comment has been added to the manuscript.
2) The presence of NdFeB particles should be shown in Figure 17.
Done
3) The labels in Figure 20 need to be placed more reasonably.
Done
4) The intrinsic coercitivity Hci of NbFeB designed by M5 should be provided such as G2 and R1000.
It not has been measured. M5 has been unsed only to obtain thge surface profile of the embedded NdFeB microstructures. VSM measurements are very time consuming. Therfore we commonly measure only G2 and R1000 to get more statistical data. Alternative measurement tools are not available since up to now such measurements on wafer level were not requested. We currenty building a dedicated tool for that within Fraunhofer.
5) The main purpose of the application should be given for the readers to better understand the work.
The introduction has been changed orrespondingly.